# Logarithmic corrections for jet production at the LHC

E. P. Byrne[1*]

**1** Higgs Centre for Theoretical Physics, University of Edinburgh,
Peter Guthrie Tait Road, Edinburgh, EH9 3FD, UK
* emmet.byrne@ed.ac.uk

February 4, 2022

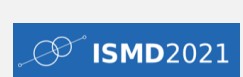

## Abstract

**Several important processes and analyses at the LHC are sensitive to higher-order perturbative corrections beyond what can currently be calculated at fixed order. One important class of logarithmic corrections are those which appear when the centre-of-mass energy of a QCD collision is much larger than the transverse momenta of the observed jets. We describe the High Energy Jets (HEJ) framework, which includes the dominant high-energy logarithms to provide all-order predictions for several LHC processes including Higgs, $W$, or $Z$ boson production in association with jets. We will summarise some recent developments, in particular the first matching of HEJ to a NLO calculation.**

## 1 Introduction

Standard Model predictions for the LHC are typically produced using general purpose Monte Carlo event generators which describe many aspects of a proton-proton collision, from PDFs to hadronisation. Here we restrict our attention to the description of the hard matrix element which describes the scattering between one parton from each proton. The hard matrix element is typically computed via fixed-order perturbation theory. For QCD amplitudes, the coupling $\alpha_s$ decreases with increasing centre of mass energy. However, there are regions of phase space where an expansion in $\alpha_s$ is unstable. For example, consider the cross section for 2→2 scattering of partons in the region where the Mandelstam variable $s$ is much larger than $|t|$. At order $\alpha_s^{n+2}$ there are terms which are of the form $\log^n(s/|t|)$. At energies accessible by the LHC the quantity $(\alpha_s \log(s/|t|))^n$ can be of order one. Logarithms of this type are called high-energy logarithms and must be summed to all orders to ensure stability of our perturbative predictions. This motivates an alternative classification of accuracy, where $N^m LL$ refers to all terms in the perturbative series of the form $\alpha_s^{n+2} \log(s/|t|)^{n-m}$.

The LL contribution to QCD amplitudes, to all orders in $\alpha_s$, takes a simple factorised form [1–3]. Rather than describing the hard matrix element of a collision at the LHC via fixed-order perturbation theory we can instead take as our starting point matrix elements which are LL accurate. High Energy Jets (HEJ) is a framework that builds on this LL accuracy. However, HEJ utilises

Monte Carlo integration for phase space integration, which means minimal approximations to the amplitude need to be made, and no formal approximations to phase space. This in turn allows HEJ to make all-order predictions for not only asymptotically large energies, but also at the scales of energy accessible by the LHC and for arbitrary cuts and analyses.

In Section 2 of we review the simple factorised form that amplitudes take in the high-energy limit. In Section 3 we show how the HEJ framework builds upon this factorised picture to obtain an improved LL accurate description, and we discuss NLL improvements to this framework. In Section 4 we discuss how this framework can be matched to existing fixed-order calculations. We review a recent application of this framework to the production of a $W$ boson in association with at least two jets before stating our conclusions.

## 2 Factorisation of amplitudes in the high-energy limit

To LL accuracy, the amplitudes for $2 \to n$ processes in QCD (with $n \geq 2$) take a simple factorised form [1–3]. These amplitudes become exact in the so-called Multi-Regge-Kinematic (MRK) limit, where for momenta $p_a, p_b \to p_1, \cdots, p_n$ the outgoing momenta are strictly ordered in rapidity $y_i$, while the transverse momenta $\mathbf{p}_i$ are of the same finite magnitude:

$$y_i \gg y_{i+1} \quad \forall i \in \{1, \cdots, n-1\}, \quad |\mathbf{p}_i| \approx |\mathbf{p}_j| \quad \forall i, j \in \{1, \cdots, n\}. \tag{1}$$

As an illustration of the simplicity of these amplitudes, we give the squared amplitude for the scattering of distinguishable quarks $q$ and $Q$ to $n$ partons:

$$\left| \mathcal{M}_{qQ \to n}^{\text{LL MRK}} \right|^2 = \frac{4s^2}{(N_C^2 - 1)} \left( \frac{g_s^2 C_F}{|\mathbf{p}_1|^2} \right) \times \left( \prod_{i=2}^{n-1} \frac{4g_s^2 C_A}{|\mathbf{p}_i|^2} \right) \times \left( \prod_{i=1}^{n-1} e^{2\alpha(\mathbf{q}_i)(y_{j+1} - y_j)} \right) \times \left( \frac{g_s^2 C_F}{|\mathbf{p}_n|^2} \right). \tag{2}$$

This expression makes full use of the simplifications allowed by the MRK region eq. 1. The only real-emission corrections which are relevant to LL order are emissions of gluons within the rapidity span of the outgoing quarks. Virtual corrections exponentiate in this limit and are described by the Regge trajectory of the associated $t$-channel momentum $\mathbf{q}_i$,

$$\alpha(\mathbf{q}_i) = -g_s^2 C_A \frac{\Gamma(1-\epsilon)}{(4\pi)^{2+\epsilon}} \frac{1}{\epsilon} \left( \frac{\mathbf{q}_i^2}{\mu^2} \right)^\epsilon. \tag{3}$$

The IR singularity here can be regularised by the procedure described in [4].

While expressions such as eq. 2 are useful for probing the asymptotic limit of QCD, the kinematic approximations are too severe to be directly applied to the LHC. This motivates constructing a framework which maintains the logarithmic accuracy of eq. 2 while relaxing many of the kinematic approximations used to obtain it.

## 3 The High Energy Jets framework

### 3.1 HEJ at LL

We begin by noting that at LO, the amplitude for $qQ \to qQ$ scattering already has the form of high-energy factorisation without taking any kinematic approximations:

$$i\mathcal{M}_{qQ \to qQ}^{(0)} = \left( ig_s T_{1a}^C j_{q \to q}^\mu(p_a, p_1) \right) \left( \frac{-i}{t} \right) \left( ig_s T_{2b}^C j_{Q \to Q\ \mu}(p_b, p_2) \right), \tag{4}$$

where we have defined the factorised spinor current

$$j_{q \to q}^{\mu}(p_1, p_a) = \bar{u}^{\lambda_1}(p_1)\gamma^{\mu}u^{\lambda_a}(p_a). \tag{5}$$

If we use these quark currents as our factorised building blocks we can construct a framework which is LO accurate for this process without loosing LL accuracy [5]. The leading-order amplitude for $qg \to qg$ contains $s$- and $u$-channel diagrams as well as a $t$-channel diagram However in ref. [6] it was shown that for the dominant helicity configurations, the $qg \to qg$ amplitude can be written exactly in the same form as $qQ \to qQ$ with $C_F$ replaced with a momentum-dependent factor $K_g$.

The HEJ form of LL amplitudes for pure-QCD processes is

$$\left|\mathcal{M}_{f_a f_b \to f_a \cdot (n-2)g \cdot f_b}^{\text{LL HEJ}}\right|^2 = \frac{1}{4(N_C^2 - 1)}\left|j_{f_a \to f_a} \cdot j_{f_b \to f_b}\right|^2 \left(g_s^2 K_{f_a}(p_a, p_1)\frac{1}{t_1}\right)\left(g_s^2 K_{f_b}(p_b, p_n)\frac{1}{t_{n-1}}\right)$$
$$\times \left(\prod_{i=1}^{n-1} e^{\omega(t_i)(y_{j+1} - y_j)}\right) \times \left(\prod_{i=2}^{n-1} \frac{-g_s^2 C_A}{t_i t_{i+1}}\left|V_g(q_i, q_{i+q})\right|^2\right), \tag{6}$$

which should be compared to eq. (2). While all-orders virtual corrections are described in the same manner, significantly more information about the emission of real radiation is contained via the HEJ form of the so-called Lipatov vertex [2,5]

$$V_g^{\rho}(q_i, q_{i+1}) = -(q_i + q_{i+1})^{\rho} + \frac{p_a^{\rho}}{2}\left(\frac{q_i^2}{p_{i+1} \cdot p_a} + \frac{p_{i+1} \cdot p_b}{p_a \cdot p_b} + \frac{p_{i+1} \cdot p_n}{p_a \cdot p_n}\right) + p_a \leftrightarrow p_1$$
$$- \frac{p_b^{\rho}}{2}\left(\frac{q_{i+1}^2}{p_{i+1} \cdot p_b} + \frac{p_{i+1} \cdot p_a}{p_b \cdot p_a} + \frac{p_{i+1} \cdot p_1}{p_b \cdot p_1}\right) - p_b \leftrightarrow p_n. \tag{7}$$

The HEJ framework can be extended to describe the production of a $W$, $Z$, or Higgs boson or same-sign $W$-pair production in association with jets [5,7–10].

## 3.2 NLL improvements to HEJ

The HEJ framework may be systematically improved by either matching to fixed-order calculations (see section 4), or by increasing the logarithmic accuracy of the all-orders amplitudes. Leading powers in $s/|t|$ in matrix elements lead to leading logarithms after integration. Regge theory gives the scaling of matrix elements in terms of the spin of the particles exchanged in the $t$-channels of the planar Feynman diagrams that can be drawn for a given process when the legs are ordered in rapidity. A flavour/momentum configuration contributes at LL if it permits an exchange of a gluon in all $t$-channels. Beyond that, a flavour/momentum configuration contributes at $N^m$LL if it permits an exchange of a gluon in all but $m$ of the $t$-channels (see ref. [11] for further details).

The first NLL component to be included in HEJ was the description of a gluon emission where that gluon was more extreme in rapidity than one of the outgoing quarks [12]. In ref. [11], the necessary pieces have been calculated to include all-order corrections to all configurations at $3j$ and above whose leading contribution is at NLL order. This also includes the potential emission of a $W$ boson from these pieces. This is a gauge-invariant subset of the full NLL correction to inclusive $2j$ or $W + 2j$ production.

## 4 NLO matching

We now describe the method used in ref. [11] to increase the fixed-order accuracy of HEJ to NLO. We write the NLO calculation of the $2j$ cross section as

$$\sigma_{2j}^{\text{NLO}} = f_{2j}^{(2)} \alpha_s^2 + \left( f_{2j}^{(3)} + f_{3j}^{(3)} \right) \alpha_s^{(3)}. \tag{8}$$

Eq. (8) is finite, after cancellation of IR singularities between the virtual corrections and unresolved real emissions. We may write the predictions of HEJ as an all-orders expression in $\alpha_s$

$$\sigma_{2j}^{\text{HEJ}} = h_{2j}^{(2)} \alpha_s^2 + (h_{2j}^{(3)} + h_{3j}^{(3)}) \alpha_s^3 + (h_{2j}^{(4)} + h_{3j}^{(4)} + h_{4j}^{(4)}) \alpha_s^4 + (h_{2j}^{(5)} + h_{3j}^{(5)} + h_{4j}^{(5)} + h_{5j}^{(5)}) \alpha_s^5 + \mathcal{O}(\alpha_s^6). \tag{9}$$

Within HEJ [13], $h_{nj}^{(n)}$ is matched to $f_{nj}^{(n)}$ where this is available. We may truncate the above series to $\alpha_s^3$, which we will refer to as HEJ@NLO:

$$\sigma_{2j}^{\text{HEJ@NLO}} = h_{2j}^{(2)} \alpha_s^2 + (h_{2j}^{(3)} + h_{3j}^{(3)}) \alpha_s^3. \tag{10}$$

In order to ensure that eq. (9) agrees with the fixed-order approach up to $\alpha_s^3$ without damaging the logarithmic accuracy of HEJ, we multiply $\sigma_{2j}^{\text{HEJ}}$ by the ratio

$$\frac{\sigma_{2j}^{\text{NLO}}}{\sigma_{2j}^{\text{HEJ@NLO}}} = 1 + (f_{2j}^{(3)} - h_{2j}^{(3)}) \prod_{n=0}^{\infty} (-1)^n \alpha_s^{(n+1)} \frac{(f_{3j}^{(3)} + h_{2j}^{(3)})^n}{(f_{2j}^{(2)})^{(n+1)}}. \tag{11}$$

In the limit of $s \gg |t|$ this ratio tends to one, where in particular, $f_{2j}^{(3)} \to h_{2j}^{(3)}$. Applying this reweighting factor to eq. (9) gives

$$\sigma_{2j}^{\text{HEJ}} \left( \frac{\sigma_{2j}^{\text{NLO}}}{\sigma_{2j}^{\text{HEJ@NLO}}} \right) = f_{2j}^{(2)} \alpha_s^2 + \left( f_{2j}^{(3)} + f_{3j}^{(3)} \right) \alpha_s^3 + \left( h_{2j}^{(4)} + h_{3j}^{(4)} + f_{4j}^{(4)} \right) \alpha_s^4 + \mathcal{O}(\alpha_s^5), \tag{12}$$

and we see that the resulting quantity is indeed NLO accurate in all of phase space.

So far we have discussed cross-sections, but the same argument holds for any particular bin in a distribution of any observable in which we might be interested. For the practical implementation of this method, we must obtain separate histograms for NLO, HEJ and HEJ@NLO. In the study ref. [11] we used Sherpa [14] with OpenLoops [15] to obtain pure NLO results. For HEJ@NLO we truncate the all-orders prediction of the relevant HEJ amplitudes. We further separate the full HEJ result into two pieces:

- The LL resummed HEJ predictions for $2j$ inclusive (in practice we observe convergence after 6 jets) including LL resummation applied to the NLL configurations discussed in section 3.2, plus $3j$ configurations which are included at LO only via fixed-order matching.

- Configurations at $\geq 4j$ which are included at LO only via fixed-order matching.

This separation allows us to avoid applying NLO-matching to the fixed-order component of the HEJ result which begins at $4j$. Finally, we can obtain the NLO-matched bin weight

$$w_{\text{HEJ2NLO}} = w_{\text{HEJ}(\geq 2j \text{ Resummed}+3j \text{ FO})} \frac{w_{\text{HEJ@NLO}}}{w_{\text{NLO}}} + w_{\text{HEJ}(\geq 4j \text{ FO})}. \tag{13}$$

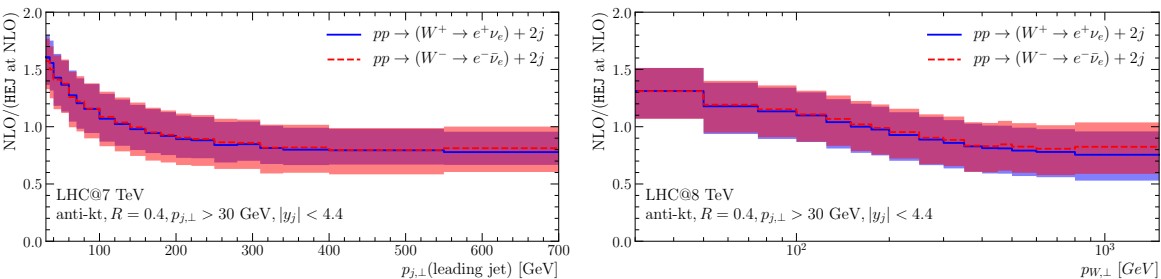

Figure 1: The NLO-matching corrections for distributions presented in [16, 17] for inclusive $W^+ + 2j$ production (blue, solid) and $W^- + 2j$ production (red, dashed).

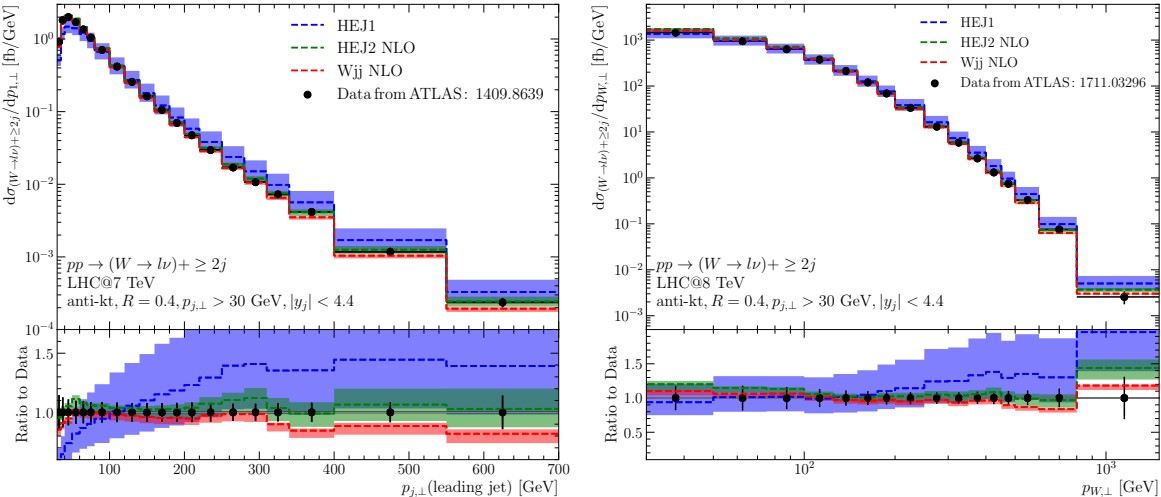

Figure 2: Predictions for $pp \to (W \to l\nu) + \geq 2j$ for the LHC, compared to ATLAS data at 7 TeV [16] and 8 TeV [17]. The content of the lines is discussed in the text.

Fig. 1 shows the matching corrections, $(w_{\text{HEJ@NLO}}/w_{\text{NLO}})$, for two distributions corresponding to measurements in refs. [16, 17]. Such $p_\perp$-based observables are of interest as they deviate from the transverse MRK condition (eq. (1)). As such, they are regions where we expect HEJ to particularly benefit from matching to NLO. We apply the matching procedure separately for $W^+$ and $W^-$ production but observe no significant difference. All calculations presented use a central scale of $\mu_r = \mu_f = H_T/2$. The scale variation bands are obtained by varying $\mu_r$ and $\mu_f$ by a factor of two around this central scale, while keeping their ratio between 0.5 and 2.

The application of these matching corrections are show in fig. 2. The blue line shows the results of the original LL HEJ framework. The green line benefits from both the inclusion of the class of NLL improvements discussed in sec. 3.2 and the application of the matching corrections shown in fig. 1. The pure NLO predictions are shown in red. We see that the NLO-matched HEJ predictions give a favourable description of the data, in particular at large $p_\perp$ where NLO fixed-order and LL corrections are both significant. Furthermore, we see that the NLO-matched HEJ predictions benefit from a significant reduction in the scale variation band.

## 5 Conclusion

In this proceeding we have introduced the High Energy Jets (HEJ) framework, which provides all-orders predictions for jet production at the LHC that include high-energy effects and restore stability to the perturbative expansion. As an example, we have applied this to the process $pp \to (W \to \ell\nu)+ \geq 2j$. We have seen that the combined effect of two improvements to the HEJ framework, namely the inclusion of a gauge invariant class of NLL improvements and the matching to a NLO calculation, leads to close agreement with data. This highlights the importance of utilising the best of fixed-order and all-order calculations in order to provide precise and stable predictions for the LHC.

## Acknowledgements

We thank the organisers for an interesting and engaging workshop. We would further like to thank the other members of the HEJ collaboration for discussions during this work. We gratefully acknowledge funding received from the ERC Starting Grant 715049 "QCDforfuture".

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
