# Peer review of "Logarithmic corrections for jet production at the LHC"

_SciPost Physics Proceedings_

## Round 1 · Referee Report · Anonymous (Referee 1) · 2022-1-31

Strengths

1- This paper is clearly written and scientifically correct 2 - The work that is reported is impressive and relevant for precision calculations of jet observables in the Regge kinematics

Weaknesses

none

Report

In this contribution the author reports on the High Energy Jets (HEJ) framework that enables a systematic resummation of a class of high energy logarithms for various LHC Boson-jet observables in the Regge kinematics. This is an interesting and impressive work that is well presented in this proceedings. In my opinion it meets the journal requirements for publication .

I 'd just have a couple of comments: - I think figure 1 is referred to after figure 2 in the text. The order of figures should be corrected in the revised version. - In figure 2, the NLO (red) curve seems to provide a similar description of the data to that of the resummed HEJ2 NLO curve. HEJ2 NLO seems to work better at high pt but I would expect the resummation to be more relevant at lower pt. Can the author comment on this?

Requested changes

  • I think figure 1 is referred to after figure 2 in the text. The order of figures should be corrected in the revised version.
  • In figure 2, the NLO (red) curve seems to provide a similar description of the data to that of the resummed HEJ2 NLO curve. HEJ2 NLO seems to work better at high pt but I would expect the resummation to be more relevant at lower pt. Can the author comment on this?

  • validity: top
  • significance: top
  • originality: top
  • clarity: high
  • formatting: excellent
  • grammar: excellent

Author:  Emmet Byrne  on 2022-02-03  [id 2153]

(in reply to Report 1 on 2022-01-31)

Thank you for your comments and your careful reading of the submission.

Regarding your two comments:
-Yes. In the revised manuscript I have maintained the order of the figures, but I have removed the initial reference to figure 2 so the figures now appear in the order they are mentioned.

-The high-energy resummation implemented by HEJ does not control evolution in pt so there is no systematic correlation between pt and the size of the high-energy corrections. This is partly compensated through the addition of NLO effects (as seen in the shape of the corrections in Fig. 1(a) for example). Similarly, the impact of resummation (including multiplicities >=4j) at large pt is absent in the pure NLO prediction which is inclined to drop below data, but this is compensated for by matching NLO and HE effects. Regarding this comment, I have modified the last two sentences of section 4.

---

## Round 2 · List of Changes

-Removed a figure reference in section 3.
-Amended the procedure for NLO reweighting in section 4.
-Minor clarification added to the description of figures in section 4.
-Minor text formatting.

You are currently on this page

Resubmission 2110.08017v2 on 7 February 2022

---

## Editorial Decision

publication_decision_taken:_accept